# Leonardo da Vinci’s Animal Anatomy: Bear and Horse Drawings Revisited

**DOI:** 10.3390/ani9070435

**Published:** 2019-07-10

**Authors:** Matilde Lombardero, María del Mar Yllera

**Affiliations:** Unit of Veterinary Anatomy and Embryology, Department of Anatomy, Animal Production and Clinical Veterinary Sciences, Faculty of Veterinary Sciences, University of Santiago de Compostela—Campus of Lugo, 27002 Lugo, Spain

**Keywords:** bear pelvic limb, dog antebrachium, horse trunk, horse and human comparative anatomy

## Abstract

**Simple Summary:**

Leonardo da Vinci was an outstanding artist of the Renaissance. He depicted numerous masterpieces and was also interested in human and animal anatomy. We focused on the anatomical drawings illustrating different parts of bear and horse bodies. Regarding Leonardo’s “bear foot” series, the drawings have previously been described as depicting a bear’s left pelvic limb; however, based on the anatomy of the *tarsus* and the *digit* (finger) arrangement, they show the right posterior limb. In addition, an unreported rough sketch of a dog/wolf *antebrachium* (forearm) has been identified and reported in detail in one of the drawings of the “bear’s foot” series. After a detailed anatomical analysis, the drawing “The viscera of a horse” has more similarities to a canine anatomy than to a horse anatomy, suggesting that it shows a dog’s trunk. Besides, the anatomies of the drawings depicting the horse pelvic limb and the human leg were analyzed from the unprecedented point of view of movement production.

**Abstract:**

Leonardo da Vinci was one of the most influencing personalities of his time, the perfect representation of the ideal Renaissance man, an expert painter, engineer and anatomist. Regarding Leonardo’s anatomical drawings, apart from human anatomy, he also depicted some animal species. This comparative study focused only on two species: Bears and horses. He produced some anatomical drawings to illustrate the dissection of “a bear’s foot” (Royal Collection Trust), previously described as “the left leg and foot of a bear”, but considering some anatomical details, we concluded that they depict the bear’s right pelvic limb. This misconception was due to the assumption that the bear’s *digit I* (1st toe) was the largest one, as in humans. We also analyzed a rough sketch (not previously reported), on the same page, and we concluded that it depicts the left *antebrachium* (forearm) and *manus* (hand) of a dog/wolf. Regarding Leonardo’s drawing representing the horse anatomy “The viscera of a horse”, the blood vessel arrangement and other anatomical structures are not consistent with the structure of the horse, but are more in accordance with the anatomy of a dog. In addition, other drawings comparing the anatomy of human leg muscles to that of horse pelvic limbs were also discussed in motion.

## 1. Introduction

Leonardo da Vinci was one of the most important renaissance personalities of his time, and the fifth centenary of his death will be commemorated in 2019. Being the illegitimate son of a notary, he did not continue the family saga and was educated privately. He had no formal education, thereby not conditioning his curiosity about the world around him. The erudite texts of his time were written in Latin and Greek, languages he did not master, and his access to the literature was therefore limited.

He was an artist and a scientist. As a painter, scientist, engineer and theorist, he produced thousands of drawings [1], personifying the ‘Renaissance man’ skilled and versed in arts and sciences [2].

His interest in anatomy was overwhelming, proven by the numerous sheets dedicated to his anatomical studies, with abundant notes and drawings, exemplifying Leonardo’s principle that anatomic parts and organs should be represented in multiple views. Considering that dissections of human corpses outside Universities were not considered appropriate by the ecclesiastical authorities, he performed some dissections of animals. According to the Royal Collection Trust [3], at the outset of Leonardo’s anatomical investigations, he was unable to procure much human material. Hence, many of his dissections were therefore of animals.

Practically his entire collection of anatomical drawings was compiled in the Windsor Codex, property of Her Majesty Queen Elizabeth II. These drawings of the human body were exhibited in an unprecedented exhibition in 2012 at the Queen’s Gallery, Buckingham Palace (London, UK). Although previous access to the collection was highly restricted, nowadays, the Royal Collection Trust offers the possibility of free access to these drawings in high resolution on its website, which greatly enables the observation of these masterpieces and their details.

Several works have been published based on these anatomical drawings, the most exhaustive ones are those from the collection of three volumes from Clark [4,5], compiling all the inventory information, the book from O’Malley and Saunders [6] and its posterior editions in 1983 and 2003 [7,8], and the official book of the exhibition. Clayton and Philo [9] and another book published in 2013 [10], the two latter reviews, mainly referred to human anatomy, although they also include comments on some animal anatomy drawings. Apart from books, there are numerous scientific articles sharing the same subject: Leonardo da Vinci’s anatomical drawings, mainly intended to some areas of expertise, such as those from Schultheiss et al. [11], Jose [12], Ganseman and Broos [13], Pasipoularides [2], Sterpetti [14], Bowen et al. [15] and West [16], among others.

It is well known that Leonardo dissected numerous animals [17]. As a result, many endeavors have been made to identify the animal of which the individual anatomical drawings have been made. In some cases, such identification is easy, while in others it is impossible [17]. Leonardo da Vinci’s methods of acquiring knowledge were observation and experiment, and for him, the study of anatomy became a science, combining both the study of structure and function [12].

Reviewing the work of several authors on the description of Leonardo’s animal anatomy drawings, and comparing them with the high-resolution images available on the website of the Royal Collection Trust [3], it can be noted that some of them were not properly described elsewhere, with some inaccuracies or misunderstandings that deserve to be discussed, probably due to the fact that the consulted authors were experts in human anatomy and, therefore, had no deep understanding of animal anatomy. Hence, it is important to point out that human anatomy could be considered similar, although with some differences, to animal anatomy. For major details, all of Leonardo da Vinci´s anatomical drawings can be accessed on the Royal Collection Trust website [3].

Regarding Leonardo’s anatomical drawings, apart from human anatomy, he also depicted some animal species such as dogs, bears, pigs, horses, oxen and monkeys. The main aim of this comparative study on anatomy was focused only on two species: Bears and horses.

According to Forlani-Tempesti [1], da Vinci mentioned bears in his notes for his anatomical treatise: “I will discourse of the hands of each animal to show in what they vary; as in the bear which has the ligatures of the toes joined above the instep”, and again: “Here is to be depicted the foot of the bear or ape or other animals to show how they vary from the foot of man or, say, the feet of certain birds” [1]. Bears are also the protagonists of other drawings of da Vinci: A bear walking (Robert Lehman Collection) and three other studies of a bear’s (or a wolf’s or dog’s) paws (1490–1495) and head (Colville Collection) [1]. However, those images of paws cannot be from a bear, simply because bears have a *manus* (hand) with five *digits* (fingers) with their *distal phalanxes* (claws) in contact with the floor. In contrast, a dog and wolf *manus* only have four *digits* ending in *unguicula* (claws) in contact to the ground, plus the *digit I* (dewclaw) medially placed at the level of the metacarpal bones.

## 2. Flaws of the Anatomy of the Bear’s *Pes* (Foot) and the Hidden *Antebrachium* (Forearm) and *Manus* from a Dog/Wolf

The set of drawings that made us realize that some inaccuracies were made in terms of their description was that of the bear’s foot (Royal Collection Inventory Number—RCIN 912372-5). Regarding RCIN 912372 (Figure 1), the first reference to it was stated by Professor William Wright in 1919 [18], in a section entitled ‘Leonardo as an Anatomist’, published by the Burlington Magazine to commemorate the Quartercentenary of Leonardo da Vinci [18] as ‘one of the finest of Leonardo’s anatomical drawings, the hind foot of a plantigrade carnivorous animal—probably a bear, a view supported by the fact that in one of the manuscripts, a reference is made to a bear’s foot’.

The reference made to this figure (RCIN 912372) by Castiglioni [19] describes it as ‘Studio di anatomia del piede umano, con artigli al posto delle unghie’, a statement that clearly evidences its misinterpretation. Previously, it was catalogued as the foot of a monster, as reported later by Clark [4]. The description of Leonardo’s bear’s foot drawings (1488–1490), made by Clayton and Philo [9] (RCIN 912372; Figure 1), indicated that “Leonardo dissected the left hind leg of a bear… The drawings show the bones, muscles and tendons of the lower leg and foot”, in accordance to O’Malley and Saunders [6], whose Comparative Anatomy Chapter states that “the drawings represent a dissection of the left leg and foot of a bear as originally pointed out by the anatomist, William Wright. There can be no question that the identification is correct”. It seems, indeed, to be a bear’s foot. In accordance, the description of the same drawing at the Royal Collection Trust website [3] states “this drawing shows with some accuracy the bones, muscles and tendons of a bear’s lower leg and foot, with the big toe, claw raised, away from the viewer”. The bear, as a plantigrade animal, walks like humans, with the whole plantar face of the *pes* (the sole with the heel) touching the ground. However, in contrast to humans, the shortest *digit* (toe) is not the fifth one (the lateral one), but the medial one, that is the first *digit* [20,21]. Hence, to the best of our knowledge, we support that the bear’s foot depicted by Leonardo corresponds to the right hind limb instead of the left one, as previously reported by O’Malley and Saunders [6] and Clayton and Philo [9] as well as at its description at the Royal Collection Trust website [3], maybe in resemblance to humans. In addition, the *calcaneus* bone of the *tarsus* is always in a lateral position to the *talus*, and the medial projection of the *calcaneus* bone to support the *talus* is quite visible, the so-called *sustentaculum tali*, with a groove to the tendon of the muscle *flexor digitalis lateralis* [22]. Figure 1 also shows, on the left-centre, a rough sketch of some muscles that continues beneath the represented bear’s *pes*. If observed upside-down (Figure 2), it seems an *antebrachium* (forearm) and *manus* of another animal. Considering all the elements depicted (muscles’ shape, the bone and the carpal orientation as well as the preliminary draft of the *manus*), we think it corresponds to a caudomedial/palmar view of the left *antebrachium* and the palmar view of a *manus* provided of short *digits* (at least *digits* shorter than those of humans, without an opposable *digit I* (thumb)). At the *carpus* level, the *retinaculum flexorum* is still patent to some extent, as a thick transverse fascial band from the medial carpal bones to the *os carpi accessorium*, forming the *canalis carpi* (canal between the proximal row of carpal bones and *retinaculum flexorum*) [22], although it has partially been removed to liberate the tendon of the muscle that seems to be the *flexor carpi radialis*, which appears to be attached at the base of the medial metacarpal bones. However, its deepest part still keeps the tendon of what seems to be the *M. flexor digitalis superficialis* at its place. This *antebrachium* and *manus* sketches are quite slim, suggesting being depicted from a dog or a wolf more than from a bear, with a more robust *antebrachium*. The proportions of the *antebrachium*/*manus* length are also more coincident with those of a dog/wolf than of a bear (with a relatively short *antebrachium*, but longer *manus*). The oblique line crossing the medial face of the radius might represent the *Vena cephalica*, which joins the *V. cephalica accessoria* (it runs dorsomedial to the *metacarpus* and *carpus*) to conform the *V. cephalica*, which continues cranially along the antebrachium. However, if that line represents the *V. cephalica*, it should have been a bit more distal (closer to the carpal joint). To the best of the authors’ knowledge, this sketch was not previously described in any of the consulted literature, maybe because the ultimate detailed representation of the bear *pes* drawing catches the entire attention of the observer.

Another one of Leonardo da Vinci’s bear *pes* drawings, RCIN 912373 recto (Figure 3), described at the Royal Collection Trust [3] as “a study of the dissection of the leg and foot of a bear, viewed in profile to the left”, corresponds to the bear’s right *pes* shown in a medial aspect (from the left). This sheet also includes an inset in the upper left, representing one *digit* viewed in profile, where the tendon of *M. flexor digitalis superficialis* fixes at two points in the second *phalanx*, between which a hole is provided, allowing the tendon of the *M. flexor digitalis profundus* to fix distally on the flexor surface of the third *phalanx* of the *pes digit*, a structure perfectly depicted by Leonardo da Vinci.

The metalpoint drawing RCIN 912374 (1488–1490; Figure 4), “a study of the dissection of the leg and foot of a bear, viewed in profile to the right”, shows the lateral side, with the perfectly defined *fibula* and three tendons associated to its distal end (*malleolus*), one in the lateral face belonging to the *M. peroneus longus* and two sliding caudal tendons of *M. peroneus brevis* and *M. extensor digitalis lateralis*. In addition, the lateral *digit* is one of the longest. Taking all this information into account, it could be concluded that the bear’s *pes* represented in this drawing is again the right one.

The last one of da Vinci’s bear *pes* drawings of this series, catalogued as RCIN 912375: “A bear’s foot” (Figure 5), shows an “outside view of the foot, partially from below” [9], where the tibia is visible and the *calcaneus* is lateral to it, corresponding also to the right bear’s *pes*, an aspect not revealed in former descriptions.

In summary, all bear’s *pes* sheets depicted by Leonardo da Vinci correspond to the right pelvic limb. However, the preliminary draft of the *antebrachium* and *manus* of a dog/wolf illustrates the left one. It is clear that Leonardo da Vinci, on many occasions, used the sheets without an order. This fact leads to the interpretation made by Keele [17] relative to Leonardo’s method of research: “he used pages of notes on particular subjects as we would use a filing system, returning to the same page at intervals of weeks, months, in some cases as long as twenty years later to record further drawings or verbal notes on the same subject”.

## 3. Anatomy of the Horse Trunk that Turned into a Dog’s Trunk

Later on, according to Clayton and Philo [10] “Writing in the mid-sixteenth century, the biographer Giorgio Vasari stated that Leonardo compiled a treatise on the anatomy of the horse. One drawing of the viscera of a large quadruped, probably a horse, does survive from this period, suggesting that Leonardo conducted full dissections to investigate the internal anatomy of the beast. But Vasari also stated that the treatise on the horse was lost when Milan was invaded by French forces in 1499. Ludovico Sforza was overthrown, and soon afterwards, Leonardo left the city and returned to Florence” [10]. This text refers to the drawing RCIN 919097-recto, entitled ‘The viscera of a horse’ (1490–1492; Figure 6), and described at the Royal Collection Trust [3] as: “an anterior view of the arteries, veins and the genito-urinary system of an animal, probably a horse,” implying that Leonardo did not name this drawing. The drawing represents the ventral aspect of the trunk of an animal (supposedly, a horse) with the lungs and the *canalis alimentarius* (esophagus, stomach and intestines) removed. The large vessels depicted at the centre, all the figure down, represent the *aorta* (on the right of the figure) and the vena *cava caudalis* (on the left of the image). The way they ramify helps us to discard the idea that this drawing represents a human being. In humans, the *aorta* ends up dividing into two branches: *Aa. iliaca commune* (*dextra* and *sinistra*). In contrast, in animals, the end of the *aorta abdominalis* (at the level of the pelvic limbs) produces two branches (*external iliac arteries*—*dextra* and *sinistra*), well depicted, and subsequently continues and produces two more (*internal iliac arteries*), well represented in the drawing, ending as the *arteria sacral median*, not depicted. Regarding the veins, the *vena cava caudalis* is formed by the confluence of two *Vv. Iliaca commune*—*dextra* and *sinistra*, each one resulting from the junction of the *V. iliaca externa* and the *V. iliaca interna*, following a similar pattern both in humans and domestic mammals.

Regarding the blood vessels, the drawing (Figure 6) provides three key points: (a) The first huge vessel (on the left of the image), reaching the heart, could be the *Vena cava cranealis*, and the other curved vessel going down is the *aorta* and its *arcus aortae*, with two big arteries leaving the aortic arch and some smaller ones (2–3) once the arch finishes and continues to the descendent aorta (*aorta descendens*). Large domestic mammals (horses—Eq, and ruminants—Ru) only have one artery deriving from the aortic arch, the *truncus brachiocephalicus*, which is then divided into a *truncus bicaroticus* (could be absent in carnivores—dogs and cats) and two *Aa. subclaviae*. In contrast, carnivores and pigs (Su) have two arteries leaving the *arcus aortae*: The *truncus brachiocephalicus* first and secondly the *A. subclavia sinistra*. (b) On the other hand, at the kidney level, there are two arteries perfectly outlined in the drawing stemming from the *aorta*: The *A. circumflexa ilium profunda* (*dextra* and *sinistra*), exclusive to carnivores [22] and dividing into the *rami craniales* and *caudales*. In contrast, the *Aa. circumflexa ilium profunda* derives from the *A. iliaca externa* in Su, Ru and Eq [22], similar to humans [23], not stemming directly from the aorta. (c) The arteria and vena *circumflexa ilium superficiales*, the first branches of the *A. femoralis* and *V. femoralis*, respectively, are exclusive to carnivores [22]. They leave their main vessels cranially oriented, at the medial and proximal part of the thigh. These vessels (a–c) in this drawing are the main clue to determine the species. Consequently, the horse representation/provenance of this drawing could be discarded. However, the horse is the unique domestic species in which the aorta does not end caudally as an *arteria sacral median*, which is not represented in the illustration.

In order to elucidate the identity of the depicted specimen, more elements were analyzed. The testicles in that position (*regio urogenitalis*-ventral pelvic region) could be mainly those of a dog. Male cats have the scrotum placed at the perineum, similar to boars, while the scrotum position of horses and bulls is more inguinal. The kidneys are not depicted as those from a horse (heart-shaped the *ren dexter* and more irregular the *ren sinister*). Those from pigs are more symmetrical and flattened, while those depicted are very similar to those from a dog. In contrast, cat kidneys have some vessels on the surface, with a radial arrangement toward the renal hilus, called capsular veins (*venae capsulares*), exclusive to cats [22], not represented in the drawing.

The hanging organ below the heart must be the liver. Accordingly, it has some very deep *incisurae interlobares* common for pigs and dogs/cats. At first glance, in between the liver and the renal arteries, it seems to be the lumbar part of the diaphragm (*crura*), but at higher magnification, it looks like an odd branch stemming from both the *aorta* and the *V. cava caudalis* (it is not clear in the drawing, maybe da Vinci had his doubts about this issue) that splits into three. There is a possibility that this vessel represents the *A. celiaca* with its three branches illustrated (*A. gastrica sinistra*, *A. hepatica* and *A. lienalis*). In that case, the organ below the heart cannot be the liver or was represented misplaced to give leadership to other, more relevant structures. Similarly, the penis has also been removed to expose the pelvic organs.

In conclusion, these details led us to confirm the hypothesis that this drawing does not represent the anatomy of a horse, as previously reported, since most of the anatomical elements are consistent with the open chest, abdomen and pelvis of a carnivore, probably a dog rather than a cat.

## 4. More on the Comparative Anatomy of Humans and Horses: The Case of the Horse and Human Anatomy of Their Pelvic Limb and Leg, Both Standing and Walking Forward

Continuing with horses, Leonardo da Vinci drew some sketches comparing the horse and human anatomy in terms of their pelvic limbs and legs, both standing and walking forward. The drawing entitled ‘The leg muscles and bones of man and horse’ (RCIN 912625; Figure 7) is described as “The muscles of a man’s legs are here studied in several views, together with the bones of the pelvis and legs, with ‘cords’ indicating the lines of action of the muscles”. At the lower centre is a diagram of the same structures in the horse, with the astute note that “to match the bone structure of a horse with that of a man you will have to draw the man on tip-toe” [3]. In 1919, and according to Wright [18], these drawings “serve to illustrate Leonardo’s methods, referred to previously, of analysing a region into its elements and of making use of comparative anatomy”. However, Wright described those illustrations as “the left hindlimb of an animal, probably a dog, and the left lower limb of a man, both drawn in the natural standing posture and both showing in the upper parts strips of corresponding muscles” [18]. Although the mentioned species of dog does not properly match with the skeleton morphology represented in this sheet, mainly due to the coxal bone (*os coxae*) shape, the presence of the third trochanter and the long metatarsal bone. Referring to these drawings, Keele [17] reported “Leonardo’s interest in movement extended from those of man to animals”, and relative to these figures, Leonardo “compares the bones of a man’s leg with those of a horse when standing, saying that to compare the two, the man must be shown standing on tiptoe. When they walk forward, this becomes even more evident”. Later on, Clayton and Philo [9] stated that “Leonardo’s study of the horse here was to some degree compromised by his knowledge of human anatomy: the pelvis is too upright and not long enough, and the femur is too long and thin”. We are in accordance with those authors: The pelvis is represented shorter than it should be, and its natural position should be a bit more horizontal. In addition, the femur should be more robust (Figure 7).

Besides, the lumbar vertebrae, which should be five or six, have large horizontal transverse processes that are not illustrated, neither are the high spinous processes present at the lumbar vertebrae and at the *os sacrum* shown, composed of five fused sacral vertebrae. In addition, the number of the lumbar and sacral vertebrae is not accurately illustrated: Three to four at the lumbar region and also at the sacrum. Moreover, the tibia and fibula are represented as bones of the horse’s leg (crus), but quite inaccurately, showing the fibula as long as the tibia, although horses only have a head and rudimentary body of the fibula (*caput* and *corpus fibulae*) [22] in a lateral position of the tibia, which barely reaches the half tibia and never articulates with the tarsus. O’Malley and Saunders [6] stated that “These three figures are for comparative purposes… The supposedly corresponding muscles of the horse are shown…but very inexactly. Presumably the cords represent the adductor, sartorius, tensor fasciae latae, gluteus superficialis or gluteus medius and gluteo-biceps of the horse”. Clayton and Philo [9] also stated that “A few muscles are represented by threads: rectus femoris from the anterior iliac spine to the patella; tensor fasciae latae from roughly the same point towards the lesser trochanter of the femur (in the horse this trochanter is much more prominent than in the human); and the gluteal muscles, represented by a number of threads (two in the horse, four in the human) running from the iliac crest towards the greater trochanter”. It is indeed the caudal part of a horse skeleton in profile because of the *os coxae* morphology, and its femur has a third trochanter (*Trochanter tertius*) specific for horses [22] (Figure 7). Hence, Clayton and Philo’s description [9] is quite inaccurate, because the lesser trochanter is placed medially with respect to the major trochanter (below the head and neck of the femur, so it cannot be seen from this lateral view), and the detail they described as lesser trochanter is, in fact, the third trochanter (*Trochanter tertius*), in which only one muscle attaches: The *M. gluteus supeficialis* [22]. On the other hand, in horses, the *M. gluteus supeficialis* should extend from the *fascia glutea* and the *os sacrum* to the *trochanter tertius*, it has a cranial head originating from the *M. tensor fasciae latae* [22]. As the unique muscle that attaches to the third trochanter is the *M. gluteus superficialis*, it is obvious that Leonardo has represented only one of its first attachments (at the *tuber coxae*, where the *M. tensor fasciae latae* comes from). Consequently, in Eq, there could be some confusion in terms of the extent of the *M. tensor fasciae latae* as it starts together with one of the heads of the *M. gluteus superficialis* (fixing at the third trochanter) to the *fasciae latae*, but in quadrupeds, never to a bone fixation as reported by Clayton and Philo [9], because it tenses the *fasciae latae*, which is one of the *M. biceps femoris* fixations.

According to Schaller [22], the *M. gluteobiceps* described by O’Malley and Saunders [6] is inexact because only Su and Ru present it (resulting from the fusion of the *M. gluteus superficialis* and the cranial portion of the *M. biceps femoris*); consequently, it does not appear in horses.

In addition, in domestic mammals, the *M. rectus femoris* (as part of the *M. quadriceps femoris*) arises from cranial to acetabulum areas [22], but not from the ventral area of the *tuber coxae* (*os ilium*), as Leonardo depicted and as Clayton and Philo stated [9]. However, it could be the *M. sartorius* as reported by O’Malley and Saunders [6], whose attachments, from the *fascia iliaca* (in Ungulates) to the medial side of the proximal portion of the tibia and *fascia cruris* [22], match Leonardo’s drawing. In humans, the *M. sartorius* runs from the *spina iliaca anterior superior* to medial to the *tuberositas tibiae* [23]. Hence, the longest thread depicted by Leonardo does not perfectly match the description. Regarding the vertical inner muscle (thread), it seems to start at the *symphysis pelvina* and to end medially and distally in the femur. This location is compatible with (a) the *M. gracilis* (flat superficial adductor of the thigh) from the *symphysis pelvina* (by *tendo symphysialis*) to the *fascia cruris* on the medial surface of the proximal portion of the crus [22], but it is placed caudal to the femur or, (b) the possibility that the *M. adductor* ‘undivided in ungulates, from the *tendo symphysialis*, *ramus caudalis ossis pubis* and *ramus ossis ischia* to the *fascies aspera*, in Eq also *condylus medialis femoris*’ [22]. In accordance to O’Malley and Saunders [6], it seems the latter (*M. adductor*) is the muscle that better fits to Leonardo’s representation. In humans, there are the *M. adductor longus* and *magnus* from near the *symphysis* and the *tuber ischiadicum* and *ramus ischii*, respectively, to the *labium mediale* of *linea aspera* both, and the *magnus* additionally to the *condylus medialis* of the femur [23]. As in a profile view from lateral, it is not possible to see the distal attachments of the represented threads; it is therefore difficult to infer which ones of the adductor muscles are depicted.

Clayton and Philo [9] also stated that “the drawing at lower right has been called an ‘anatomical fantasy’, blending the bones of a horse with those of a man. It is much more likely that Leonardo intended it to be purely human, but incorporated errors (in particular, the extended ischium below the coccyx; cf. no. 64b, in which the error is corrected), derived from his superior knowledge at that date, of equine anatomy-the study of human bones and muscle threads to the left display the same errors”. Should it say: … because of his superior knowledge of human anatomy. O’Malley and Saunders [6] referred to this figure as ‘the elongation of the innominate bone and the length of the coccyx suggest the pelvis of an animal rather than of a man. In addition, the observer will note a trochanter tertius below the greater trochanter. From these appearances, this figure seems to have been derived by the expansion of animal bones to the approximate proportions of the human’. Regarding RCIN 919012 recto: The skeleton c.1510–1511 (figure not included), drawn 2 to 5 years later than RCIN 912625 recto (Figure 7), Clayton and Philo [9] stated that “Leonardo has here corrected the length of the ischium”, referring to the lower right figure. According to Wright [18], these drawings “serve to illustrate Leonardo’s methods, referred to previously, of analyzing a region into its elements and of making use of comparative anatomy”.

There are other illustrations depicting the comparative anatomy between the man and the horse, belonging to the ‘Manuscrit K’ from the Bibliothèque de l’Institut de France [24]. One of them, folio 102 recto (Figure 8), compares the right horse hind limb (in a lateral view) with the right human leg in a frontal view. The drawing at the left centre (Figure 8B) is without doubt the laterocaudal view of the right pelvic limb of the horse, and again he has drawn a fibula that extends much too far distally along the caudal aspect of the tibia for the horse (this fact adds to the evidence that the limb depicted in Figure 7 is indeed an equine limb).

Nevertheless, the most impressive drawing in ‘Manuscrit K’ about this issue corresponds to Folio 109 verso (Figure 9), in which both sketches are reported to compare a man’s leg with a horse´s hind limb when walking forward, for which the man should be on tiptoes [17]. The illustration shows part of the trunk and the left leg of a man seen in left profile and compared with the left posterior limb of a horse, but not keeping a relative size/proportion.

Here, however, it is not clear that the represented man is walking forward, because the flexed knee during walking is concurrent with tiptoe contact to the ground, but not the trunk inclination/angle as depicted. This position is similar to that when a man or a horse is taking impulse from below (flexing their legs or hind limbs) to be able to jump. These illustrations exemplify what O’Malley [7] stated about Leonardo’s interests: “they are directed towards the structure of the body in relationship to its workings”. Maybe Leonardo tried to show a parallelism between the relative positions of bones and muscles of the horse, captured and transferred to a human skeleton in order to describe and understand the combination of the flexion of the joints together with the muscle action to accumulate the energy needed on the impulse of jumping, as it is described in the text of the following page of the manuscript K [24] (Folio 110 (30) recto: ‘saut de l’homme’; compare Figure 7 on standing position vs. Figure 9 on flexion; Figure 8 seems to be a previous sketch to Figure 9).

The main difference between the drawing at the Royal Collection Trust [3] (Figure 7) and those from ‘Manuscrit K’ [24] (Figure 8 and Figure 9 ) is that the muscle is represented as an inverted ‘V’, illustrated as different threads in the Royal Collection Trust. The inverted ‘V’ muscle has its apex at the *tuber coxae*, and the shorter part fixes laterally at the third trochanter. The longest part goes to the *condylus medialis* of the tibia in one of them (Figure 9), but it is not clear in the other folio (Figure 8) because its fixation should have been out of the sheet (lower sketch) or is not drawn with enough detail (upper sketch), indicating a rough study previous to Figure 9. In addition, just analyzing the depicted muscles and threads of these illustrations, it is clear that in Manuscript K [24] (Figure 8 and Figure 9), the muscles are represented as a wide thread with a sort of ‘belly’. However, in the drawing RCIN 912625 (Figure 7), slimmer threads are depicted; hence, it could be deduced that the sketches from Manuscript K were made previously to the drawing of the Royal Collection Trust, just as an evolving progress in his studies, a trend in all artists´ lives, as a result of a lifelong development in their conceptualization and abstraction ability/aptitude. However, comparing the contents of both illustrations (Figure 7 and Figure 9), it is not likely that Leonardo started to study the flexed hind limb to approach its standing position later. These facts should be taken into account when dating back to Leonardo’s work, as those sheets are assigned more or less to the same periods of c.1506–1508 (Figure 7) and c.1503–1508 (Figure 8 and Figure 9).

## 5. Conclusions

Leonardo da Vinci was an outstanding artist and scientist, looking for the reasons and mechanisms of all the aspects he had studied during his entire life. He tried to find the common clues shared by human and animal anatomy through comparative anatomy. He produced an impressive collection of anatomical drawings, which were, in general, quite accurate. However, some later studies have misunderstood some of them. Hence, a deep anatomical insight into the bear and horse anatomical drawing collections revealed some inaccuracies, which, for some anatomical elements, we identified and amended. Such is the case of the depicted ‘bear’s foot’ series, described as the left hind limb, but based on their *digits* and *tarsus*, it is the right one. Regarding the horse anatomy, one of Leonardo’s drawings illustrating the internal and dorsal parts of the horse trunk, if considered the different blood vessels depicted and other viscera, seems to be from a carnivore (probably a dog) rather than that of a horse, as previously reported. Other drawings illustrating the comparative anatomy of the horse hind limb and the human leg were also reconsidered in a new approach, assuming that Leonardo da Vinci used the comparative anatomy in order to understand the process to produce movement, especially when jumping.

## Figures and Tables

**Figure 1 animals-09-00435-f001:**
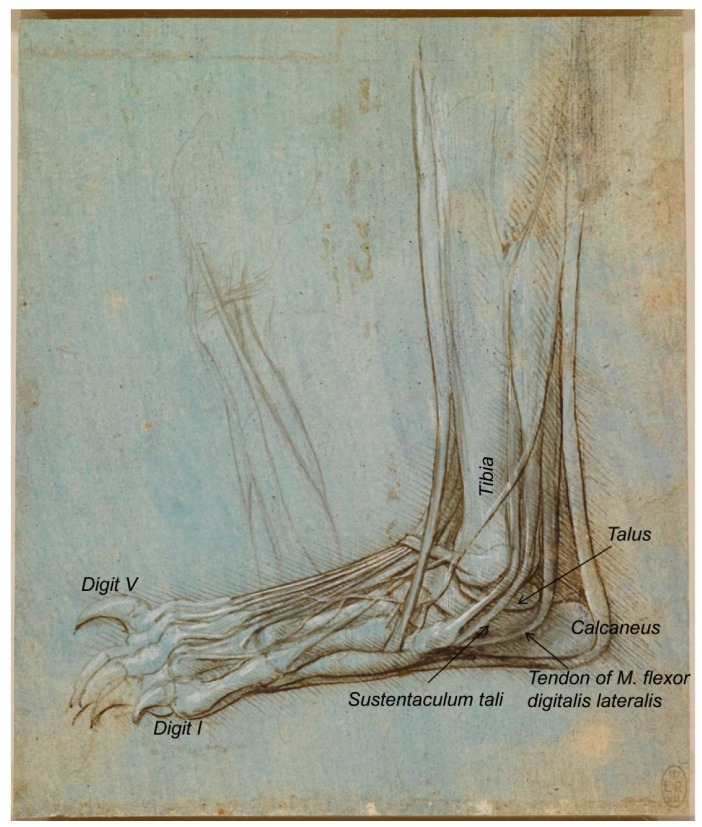
Bear’s foot series—Number 1. Bear distal right pelvic limb/*pes*, medial aspect. A bear’s foot c.1488–1490. Modified from www.rct.uk/collection/912372 (Royal Collection Trust [3]). This image is credited as Royal Collection Trust/© Her Majesty Queen Elizabeth II 2019.

**Figure 2 animals-09-00435-f002:**
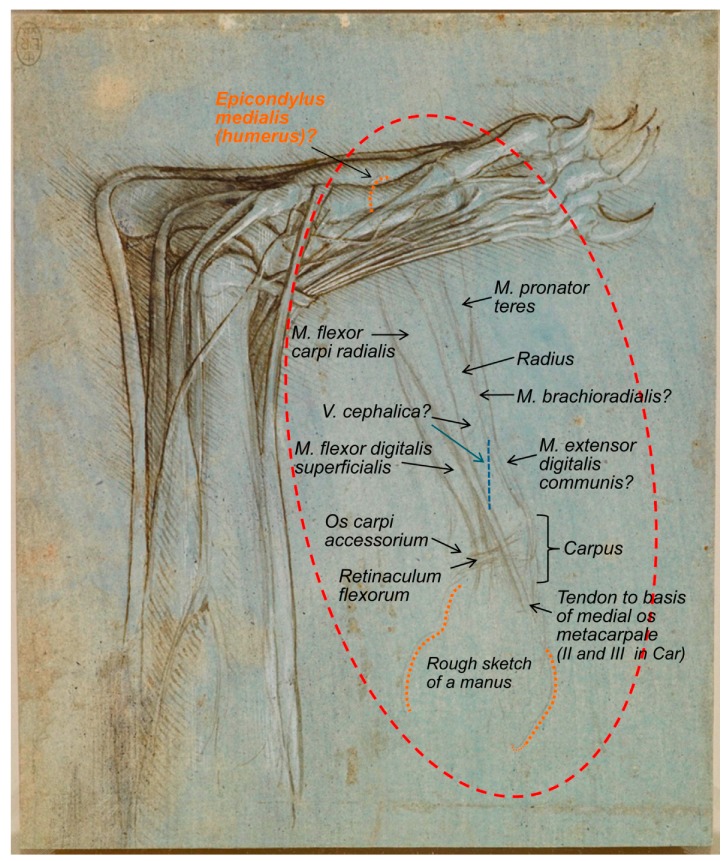
An *antebrachium* of a dog/wolf. Dog/wolf *antebrachium* and *manus*, caudomedial/palmar view. A bear’s foot c.1488–1490. Modified from www.rct.uk/collection/912372 (Royal Collection Trust [3]). This image is credited as Royal Collection Trust/© Her Majesty Queen Elizabeth II 2019.

**Figure 3 animals-09-00435-f003:**
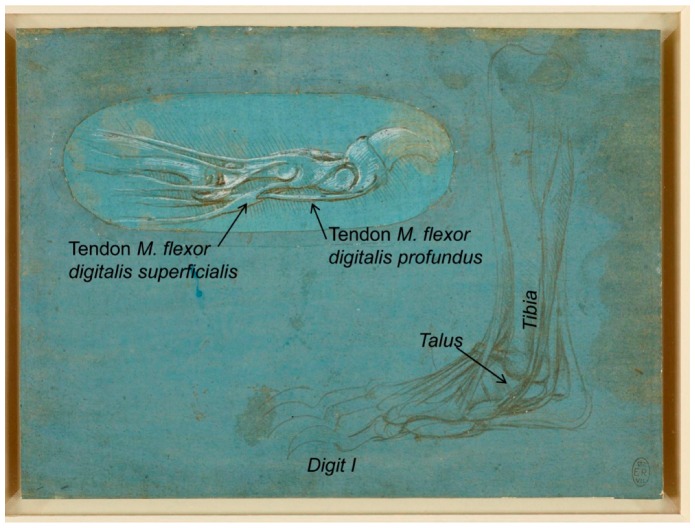
Bear’s foot series—Number 2. Upper left: Bear *pes digit*. Down right: Bear distal right pelvic limb/*pes*, medial aspect. A bear’s foot c.1488–1490. Modified from www.rct.uk/collection/912373 Recto (Royal Collection Trust [3]). This image is credited as Royal Collection Trust/© Her Majesty Queen Elizabeth II 2019.

**Figure 4 animals-09-00435-f004:**
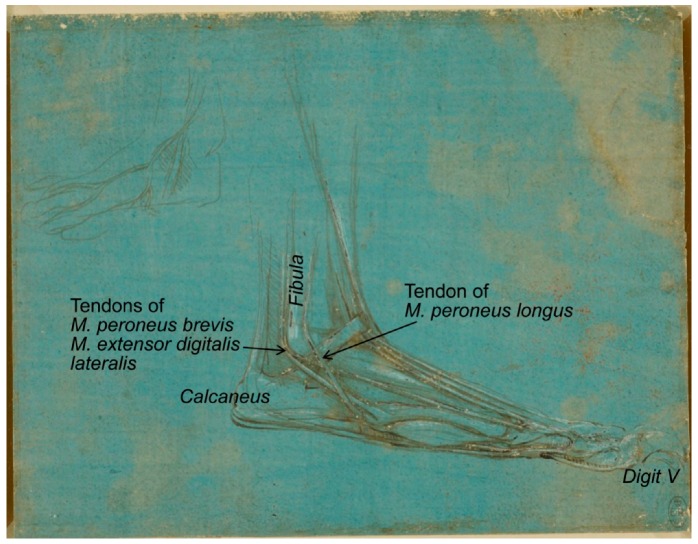
Bear’s foot series—Number 3. Bear distal right pelvic limb/*pes*, lateral view. A bear’s foot c.1488–1490. Modified from www.rct.uk/collection/912374 (Royal Collection Trust [3]). This image is credited as Royal Collection Trust/© Her Majesty Queen Elizabeth II 2019.

**Figure 5 animals-09-00435-f005:**
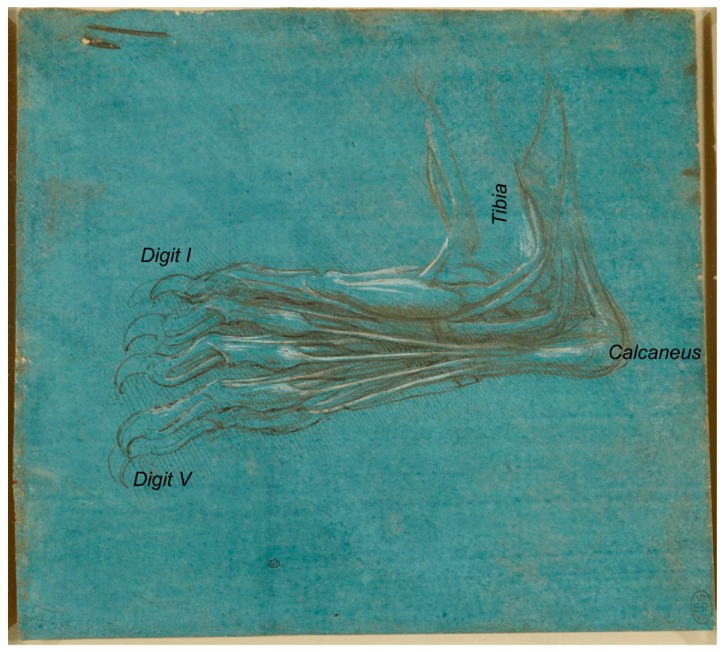
Bear’s foot series—Number 4. Bear distal right pelvic limb/*pes*, plantaromedial oblique view. A bear’s foot c.1488–1490. Modified from www.rct.uk/collection/912375 (Royal Collection Trust [3]). This image is credited as Royal Collection Trust/© Her Majesty Queen Elizabeth II 2019.

**Figure 6 animals-09-00435-f006:**
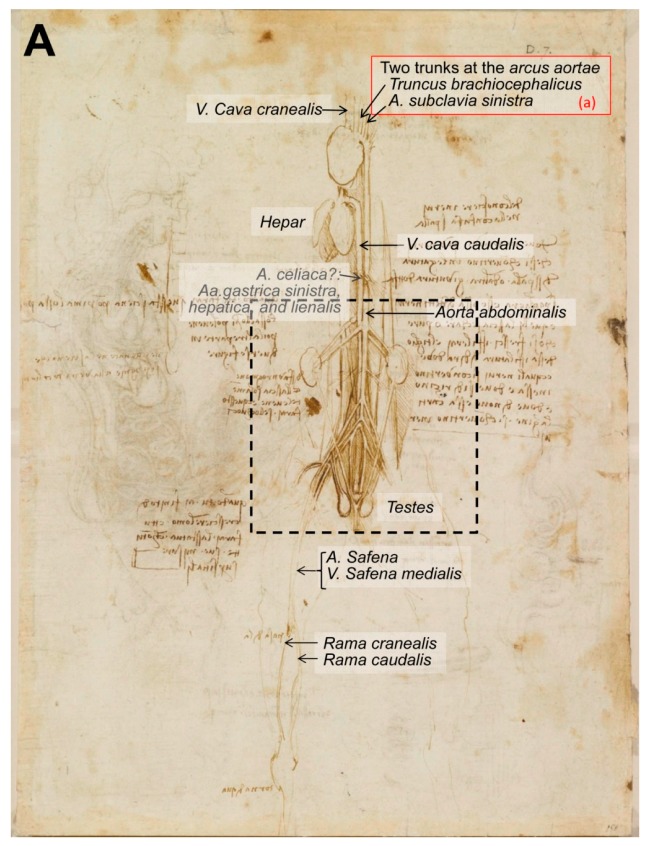
The viscera of a horse? (**A**) Whole drawing representing the ventral aspect of the trunk of an animal with the *canalis alimentarius* and lungs removed. (**B**) Inset at higher magnification depicting the lumbar and pelvic regions. The viscera of a horse c.1490–1492. Modified from www.rct.uk/collection/919097 recto (Royal Collection Trust [3]). This image is credited as Royal Collection Trust/© Her Majesty Queen Elizabeth II 2019.

**Figure 7 animals-09-00435-f007:**
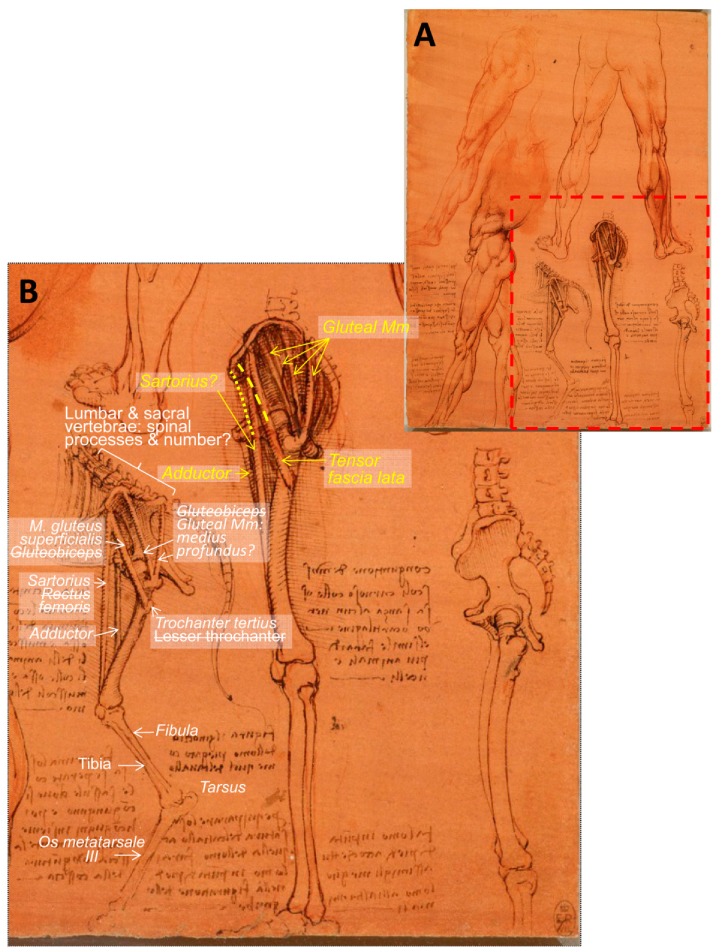
Series of comparative anatomy of man’s leg and horse’s pelvic limb—Number 1. (**A**) Whole drawing. (**B**) Inset at higher magnification showing a horse left pelvic limb (on the left) and a human left leg (centre and right), lateral aspect. The leg muscles and bones of man and horse c.1506–1508. Modified from www.rct.uk/collection/912625 (Royal Collection Trust [3]). This image is credited as Royal Collection Trust/© Her Majesty Queen Elizabeth II 2019.

**Figure 8 animals-09-00435-f008:**
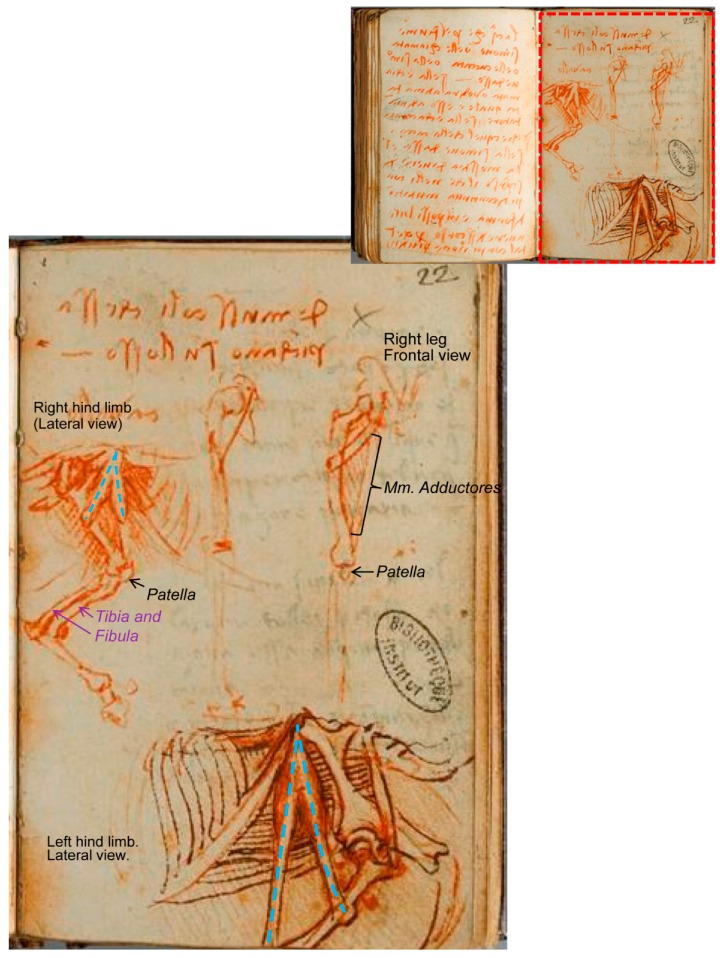
Series of the comparative anatomy of a man’s leg and horse’s pelvic limb—Number 2. (**A**) Two pages. (**B**) Inset: Higher magnification of Folio 102(22) recto: ‘Anatomie, cheval’. Modified from Ms 2181-folio 102(22)-manuscrit K (Bibliothèque de l’Institut de France [24]) (1503–1508). This image presentation is authorized in limited time period by Réunion des Musées Nationaux-Grand Palais (R.M.N.-Grand Palais), and the Bibliothèque de L’Institut de France. Photo © RMN-Grand Palais (Institut de France)/René-Gabriel Ojéda.

**Figure 9 animals-09-00435-f009:**
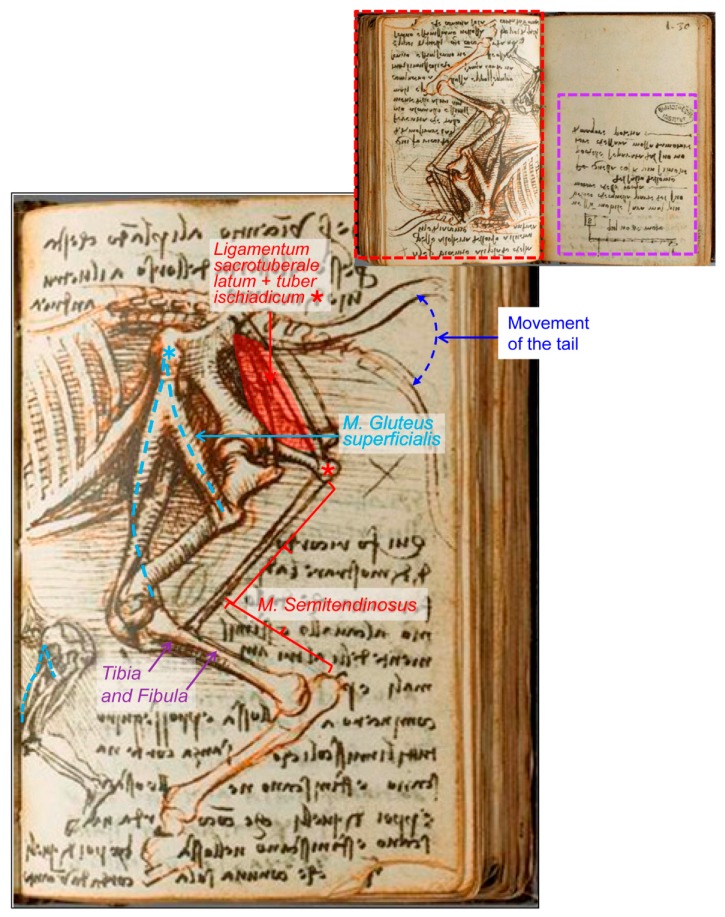
Series of the comparative anatomy of a man’s leg and horse’s pelvic limb—Number 3. (**A**) Two pages: Folio 109 verso (red) and folio 110 recto (violet inset with da Vinci notes about ‘Leviers, movements, saut de l’homme’). (**B**) Folio 109 verso: ‘Anatomie comparée de l’homme et des animaux (chevaux)’. Red inset upside down at higher magnification displaying the comparative between the man left leg (bottom, left) and the horse left pelvic limb, lateral aspect. Modified from Ms 2181-folio 109(29-30) and 110 (30)-manuscrit K (Bibliothèque de l’Institut de France [24]) (1503–1508). This image presentation is authorized in limited time period by Réunion des Musées Nationaux-Grand Palais (R.M.N.-Grand Palais), and the Bibliothèque de L’Institut de France. Photo © RMN-Grand Palais (Institut de France)/René-Gabriel Ojéda.

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
