# Peer review of "Leonardo da Vinci’s Animal Anatomy: Bear and Horse Drawings Revisited"

_animals, 2019, doi:10.3390/ani9070435_

Round 1

Reviewer 1 Report

This year, as the fifth centenary of Leonardo da Vinci passment is commemorated all around the world. Leonardo da Vinci was an expert painter, engineer and anatomist. He made many anatomical drawings, mainly representing human anatomy, although he also studied animal anatomy, but he did not describe them. The authors compared different drawings with descriptions given by non-veterinary specialists, and they found some misunderstandings of what was represented. This is the right time to amend all those misinterpretations of those anatomical drawings representing bears and horses. Moreover, a new animal sketch was revealed. These amendments will highlight anatomical drawings by Leonardo on animal anatomy for wider audience, such as anatomists, veterinary academics, researchers, and those interested in the world of the ‘Renaissance man’ skilled and versed in arts and sciences.

The authors used comparative anatomy to stand out the knowledge on animal anatomy in the Renaissance, specifically five hundred years back. The manuscript amends some descriptions and reveals new undescribed features. Those findings honour the excellence of Leonardo da Vinci, not only as an artist, but also as an anatomist, even on the lest known field of animal science. There are no other studies on this specific field since the first descriptions by O´Malley & Saundersand Clark on years 1952 and 1968-1969, respectively. Leonardo drawings are visited several times at the Royal Collection Trust website; therefore, it is necessary an accurate description of their illustrations, specifically on less known animal anatomy.

The claims are novel, because it includes the amendments, and a new drawing has been revealed. Many published papers compromise novelty. I do believe that this is a complete and final study on the specific anatomical work on bears and dogs by Leonardo, of broad interest for academics, researchers and the general public interested in Leonardo as an artist and a anatomist. authors have referenced all of them needed to prove the novelty of their findings.

The claims are also convincing. All the results were properly detailed and discussed and based on anatomical evidences. The amendments and new findings completely describe the Leonardo’s studies on animal anatomy. This manuscript strengthens the Leonardo’s skills and knowledge on animal anatomy. I think this is a complete and final study on the specific anatomical work on bears and horses by Leonardo, of broad interest for academics, researchers and the general public interested in Leonardo as artist and anatomist.

Minor comments:

My only minor suggestion is that the drawings are too small and do not look good. It would be better to provide sharper images.

Author Response

This year, as the fifth centenary of Leonardo da Vinci passment is commemorated all around the world. Leonardo da Vinci was an expert painter, engineer and anatomist. He made many anatomical drawings, mainly representing human anatomy, although he also studied animal anatomy, but he did not describe them. The authors compared different drawings with descriptions given by non-veterinary specialists, and they found some misunderstandings of what was represented. This is the right time to amend all those misinterpretations of those anatomical drawings representing bears and horses. Moreover, a new animal sketch was revealed. These amendments will highlight anatomical drawings by Leonardo on animal anatomy for wider audience, such as anatomists, veterinary academics, researchers, and those interested in the world of the ‘Renaissance man’ skilled and versed in arts and sciences.

The authors used comparative anatomy to stand out the knowledge on animal anatomy in the Renaissance, specifically five hundred years back. The manuscript amends some descriptions and reveals new undescribed features. Those findings honour the excellence of Leonardo da Vinci, not only as an artist, but also as an anatomist, even on the lest known field of animal science. There are no other studies on this specific field since the first descriptions by O´Malley & Saundersand Clark on years 1952 and 1968-1969, respectively. Leonardo drawings are visited several times at the Royal Collection Trust website; therefore, it is necessary an accurate description of their illustrations, specifically on less known animal anatomy.

The claims are novel, because it includes the amendments, and a new drawing has been revealed. Many published papers compromise novelty. I do believe that this is a complete and final study on the specific anatomical work on bears and dogs by Leonardo, of broad interest for academics, researchers and the general public interested in Leonardo as an artist and a anatomist. authors have referenced all of them needed to prove the novelty of their findings.

The claims are also convincing. All the results were properly detailed and discussed and based on anatomical evidences. The amendments and new findings completely describe the Leonardo’s studies on animal anatomy. This manuscript strengthens the Leonardo’s skills and knowledge on animal anatomy. I think this is a complete and final study on the specific anatomical work on bears and horses by Leonardo, of broad interest for academics, researchers and the general public interested in Leonardo as artist and anatomist.

Minor comments:

My only minor suggestion is that the drawings are too small and do not look good. It would be better to provide sharper images.

ANSWER

We are very grateful for your supportive and constructive comments about our manuscript. We do think it is interesting to review some the masterpieces of Leonardo da Vinci, some of them not properly described, and consequently, inaccurately catalogued.

Relative to your suggestion of changing the small images by other sharper, we have included new images (all of them 15 cm width) that are more intelligible to follow the descriptions in the text.

In addition, the figure 6 (‘the trunk of a horse?’) was divided into two figures: figure 6A and 6B, to better show all the details.

However, and as it was suggested by the Reviewer-2 to avoid confusion, some of the labels were changed to follow the veterinary anatomical nomenclature.

We do hope our revised manuscript meets your approval.

Reviewer 2 Report

This is a very interesting body of work taking a fresh look at some of Leonardo Da Vinci’s anatomical drawings shedding new light on his work as well as correcting some earlier misconceptions with regards to the anatomy depicted. This will be of interest to anyone with an interest in the art of anatomy, and highlights the value of comparative anatomists studying these drawings.

General comments

Terminology: The manuscript could benefit from using anatomical terminology more consistently. When comparing across species this aids overall accuracy, and avoid misunderstanding/confusion (which is what this paper is trying to clear up!). For example, instead of ‘first toe’ use digit I. If you use the term digit (which would be an anatomically correct term for all animals) you remove any ambiguity.

Although I comment on the wording used in the Simple Summary below, overall I did not find the English a major issue.

Simple Summary

Some slightly awkward English here, could benefit from some minor English corrections, some terms that just don’t make sense i.e. line 14 …they outcame…

Abstract

No comments

1. Introduction

No issues with the introduction, other than terminology. It is a mixture of paws/hands/fingers/claws. Again, as in the general comments, this can lead to confusion.

Line 91 page 2: The term manus rather than hand should be used. It makes no sense talking of bears having hands with fingers. To keep the manuscript accessible, it would be perfectly acceptable to put the lay terms in brackets after the proper anatomical term or provide explanatory footnotes if the journal prefers this option.  Likewise, dewclaw should have the term digit I in brackets after it or vice versa.  

2.

The title should use proper anatomical terms. The terms bear’s foot immediately causes confusion due to the bear having four feet in lay terms. Which one are we considering here, manus or pes? As the previous section ended up discussing the manus, we are now suddenly asked to consider the pes. Clarification required, and this is best achieved by the standard anatomical terms.

Figure 1: I am perfectly comfortable with the argument that we are looking at the pes of the right pelvic limb of a bear here. Your anatomical references are spot on, both bones and muscles/tendons as well as the digits show this is a medial view which means this has to the right limb.

Figure 2: Labelling of Figure: I mostly agree with this, and I am comfortable with the description in the text (although refer to previous comments on terminology). The Figure has been labelled and I think that needs some further considerations. Firstly, the view is a caudomedial view of the left antebrachium, highly likely to be of a canine/dog as described in the text. The Figure labels should also describe it as caudomedial/palmar view of antebrachium/manus.

I am not convinced you can see M. extensor carpi radialis on this view (it can be hard to discern on a medial view, let alone a caudomedial view), I think this is M. pronator teres, as I cannot see any convincing outline of M. pronator teres where you have labelled it. It is usually a very distinct muscle belly. This might be due to inferior quality of my image, but I think we might be seeing some of M. pronator quadratus or head of another deep muscle here instead.

I cannot make out the manus (hand) so I am unable to comment on the distal aspects and will have to take them as described.

Figure 3: I agree, medial aspect of R pelvic limb.

Figure 4: Agree, lateral aspect of R pelvic limb.

Figure 5: Terms like ‘outside view’ is not helpful, it is meaningless. I would describe this as a plantaromedial oblique view of distal right pelvic limb/pes.

3.

I think it should be specified that we are looking at the ventral aspect of the trunk of an animal with the intestines and lungs removed.

Page 7 line 184, 188 – I think the term viscera (organ) is inaccurate here, as the digestive viscera and the lungs have been removed.

Figure 6: I agree with the argument animal vs. human. (a) Agree this depicts carnivore/porcine aortic arch anatomy with a left subclavian artery. (b) Agree (c) Agree.

The anatomy points towards a carnivore, probably a dog rather than a cat.

4.

Page 10, line 287: Dogs do have a third trochanter on the lateral aspect of the femur where the superficial gluteus muscle inserts as in this Figure 7. Its origin in the dog is from the sacrum and caudal vertebrae, not the coxae as depicted here. It is incorrect however to say the third trochanter is specific to the horse. I do however agree with the authors’ comments with regards to Clayton & Philo’s descriptions line 288 onwards.

Page 11 line 332 – confusing to refer to RCIN 919012 as not shown in any Figures?

An additional interesting detail in Figure 8 worthy of comment is that this is without doubt the laterocaudal view of the right pelvic limb of the horse, and again he has drawn a fibula that extends much too far distally along the caudal aspect of the tibia for the horse (adds to the evidence that the limb depicted in Figure 7 is indeed an equine limb).

5. Conclusions

No further comments.

Author Response

This is a very interesting body of work taking a fresh look at some of Leonardo Da Vinci’s anatomical drawings shedding new light on his work as well as correcting some earlier misconceptions with regards to the anatomy depicted. This will be of interest to anyone with an interest in the art of anatomy, and highlights the value of comparative anatomists studying these drawings.

Response: We appreciate your encouraging comments.

General comments

Terminology: The manuscript could benefit from using anatomical terminology more consistently. When comparing across species this aids overall accuracy, and avoid misunderstanding/confusion (which is what this paper is trying to clear up!). For example, instead of ‘first toe’ use digit I. If you use the term digit (which would be an anatomically correct term for all animals) you remove any ambiguity.

Response: All the terms that could generate some confusion in the main body of the manuscript have been changed following the veterinary anatomical nomenclature, except in the title of the figures, because we prefer keeping the original name of the Royal Collection Trust inventory for further references. However, labels and figure captions are properly described, using the veterinary anatomical nomenclature.

Although I comment on the wording used in the Simple Summary below, overall I did not find the English a major issue.

Response: As the Editor and other Reviewers have pointed out, the simple summary and the abstract have been proofreading by an English native speaker.

Simple Summary

Some slightly awkward English here, could benefit from some minor English corrections, some terms that just don’t make sense i.e. line 14 …they outcame…

Response: After an English native proofreading, we think the short summary and the abstract are correct now.

Abstract

No comments

Response: OK

1. Introduction

No issues with the introduction, other than terminology. It is a mixture of paws/hands/fingers/claws. Again, as in the general comments, this can lead to confusion.

Line 91 page 2: The term manus rather than hand should be used. It makes no sense talking of bears having hands with fingers. To keep the manuscript accessible, it would be perfectly acceptable to put the lay terms in brackets after the proper anatomical term or provide explanatory footnotes if the journal prefers this option. Likewise, dewclaw should have the term digit I in brackets after it or vice versa.

Response: The lay terms have been changed and the standard veterinary anatomical terminology was used instead, leaving the non-technical word in brackets the first time the anatomical term appeared, starting at Line 144 and onwards (tracked version) or Line 93 and onwards (clean version).

2.

The title should use proper anatomical terms. The terms bear’s foot immediately causes confusion due to the bear having four feet in lay terms.

Which one are we considering here, manus or pes? As the previous section ended up discussing the manus, we are now suddenly asked to consider the pes. Clarification required, and this is best achieved by the standard anatomical terms.

Response: As commented in the introduction the lay names were changed by anatomical nomenclature, unless in the names of the figures (just for further references), using the standard anatomical nomenclature to describe them at the figure captions.

Figure 1: I am perfectly comfortable with the argument that we are looking at the pes of the right pelvic limb of a bear here. Your anatomical references are spot on, both bones and muscles/tendons as well as the digits show this is a medial view which means this has to the right limb.

Response: OK

Figure 2: Labelling of Figure: I mostly agree with this, and I am comfortable with the description in the text (although refer to previous comments on terminology). The Figure has been labelled and I think that needs some further considerations. Firstly, the view is a caudomedial view of the left antebrachium, highly likely to be of a canine/dog as described in the text. The Figure labels should also describe it as caudomedial/palmar view of antebrachium/manus.

I am not convinced you can see M. extensor carpi radialis on this view (it can be hard to discern on a medial view, let alone a caudomedial view), I think this is M. pronator teres, as I cannot see any convincing outline of M. pronator teres where you have labelled it. It is usually a very distinct muscle belly. This might be due to inferior quality of my image, but I think we might be seeing some of M. pronator quadratus or head of another deep muscle here instead.

I cannot make out the manus (hand) so I am unable to comment on the distal aspects and will have to take them as described.

Response: The name of M. pronator teres has been changed to the right of the figure, being removed from the left and leaving that outline without any label (it is not easy to speculate which structure is).

The manus is shown as a very incipient sketch and very difficult to show its limits (they are outlined in orange). Just because of the shape we deduce it is from a carnivore (dog/wolf).

Figure 3: I agree, medial aspect of R pelvic limb.

Response: OK.

Figure 4: Agree, lateral aspect of R pelvic limb.

Response: OK.

Figure 5: Terms like ‘outside view’ is not helpful, it is meaningless. I would describe this as a plantaromedial oblique view of distal right pelvic limb/pes.

Response: The sentence in line 168 (Line 264, tracked version; Line 174, clean version), …shows an ‘outside view of the foot, partially from below’ are the exact words used by Clayton and Philo [9] (page 44); therefore, then we decided to keep them but with quotation marks.

The description a ‘plantaromedial oblique view of distal right pelvic limb/pes’ you indicate is included in the figure 5 caption.

3.

I think it should be specified that we are looking at the ventral aspect of the trunk of an animal with the intestines and lungs removed.

Response: We accept your suggestion and this information has been included in the text (page 8, lines 293-294, tracked version; Lines 197-198, clean version), as well as at the legend of Figure 6. However, we decided to use ‘canalis alimentarius’, instead of ‘intestines’, because the esophagus and stomach had also been removed. The description that appears at the Royal Collection Trust website [3] was also included (Lines 291-292, tracked version; Lines 195-196, clean version), and a brief comment about the authorship of the title (suggested by the Reviewer-3).

Previously we have removed the sentence (Lines 189-90): ’Although at a glance, it could appear to be a human figure, if we have a close look at its legs, their outline denotes they belong to an animal’ because it does not give extra information and Reviewer-3 said it was too evident that the drawing does not represent a man because of the liver.

Page 7 line 184, 188 – I think the term viscera (organ) is inaccurate here, as the digestive viscera and the lungs have been removed.

Response: As these lines 184-188 are a citation from Clayton and Philo [10] and the text goes between quotation marks (lines 286-290 tracked version; lines 190-194, clean version), we prefer not to change the words used to describe the drawing.

Figure 6: I agree with the argument animal vs. human. (a) Agree this depicts carnivore/porcine aortic arch anatomy with a left subclavian artery. (b) Agree (c) Agree.

The anatomy points towards a carnivore, probably a dog rather than a cat.

Response: We are glad and fully agree. This latter remark was also included at the end of chapter 3 (line 381, tracked version; line 254, clean version).

4.

Page 10, line 287: Dogs do have a third trochanter on the lateral aspect of the femur where the superficial gluteus muscle inserts as in this Figure 7. Its origin in the dog is from the sacrum and caudal vertebrae, not the coxae as depicted here. It is incorrect however to say the third trochanter is specific to the horse. I do however agree with the authors’ comments with regards to Clayton & Philo’s descriptions line 288 onwards.

Response: We try to follow the Veterinary Anatomical Nomenclature [22] as a reference, and relative to trochanter tertius (page 66) it is said: ‘third trochanter in eq, lateral protuberance for attachment of M. gluteus superficialis.” In addition, if it is considered the attachment place of the M. gluteus superficialis [22] (on its page 124) it says: ”M. gluteus superficialis. On caudal surface of M. gluteus medius. From Fascia glutea and Os sacrum to Tuberositas glutea (Car) or Trochanter tertius (eq)…’ Hence, in the carnivores there is a small protuberance placed distal to the trochanter major, but never as manifest as trochanter tertius in horses. Another domestic species with a manifest trochanter tertius is the rabbit, but other elements in this drawing are not in accordance with this species, such as the long tail, pes anatomy and pelvic limb posture.

Consequently, we think we should keep this sentence as it is.

Page 11 line 332 – confusing to refer to RCIN 919012 as not shown in any Figures?

Response: We do not think we should include this drawing “the skeleton” (shown below) because is referred only to the human skeleton. However, we have made reference to it because it is dated by 1510-11, only 2-5 years later than Figure 7 and an evolution is shown in the coxal bone, “correcting the length of the ischium”. Therefore, it is clear that Leonardo da Vinci was evolving as an artist, improving his drawings in parallel to his learning in anatomy, and producing more accurate anatomical studies.

We added:  ‘(figure not included)’ at Line 488 (tracked version) or Line 347 (clean version). However, if you consider interesting include an additional figure, we could do it, as supplemental information.

Other option would be removing the whole sentence including that reference (RCIN 919112) from line 332-336 (lines 488-492, tracked version; lines 347-351, clean version).

An additional interesting detail in Figure 8 worthy of comment is that this is without doubt the laterocaudal view of the right pelvic limb of the horse, and again he has drawn a fibula that extends much too far distally along the caudal aspect of the tibia for the horse (adds to the evidence that the limb depicted in Figure 7 is indeed an equine limb).

Response: As suggested, we have included this information about the fibula in the text referred to Figure 8 (Lines 498-501, tracked version; Lines 355-358, clean version).

5. Conclusions

No further comments.

Response: We are very grateful for your wise and helpful comments. We hope you consider that our manuscript has improved and it is now acceptable for publication.

Reviewer 3 Report

This manuscript is a detailed re-evaluation of the historical interpretations of a few of Leonardo da Vinci’s illustrations. I can understand the interest in a historical figure like da Vinci. However, it was not da Vinci who misidentified the bear’s foot as left rather than right, it was the interpretation of others centuries later. Since this is not a reflection on da Vinci but on others, then I wonder how significant this is even from a historical perspective. The instances of inaccuracies by prominent figures in age-old scientific literature are an innumerable. Unless they are misguiding current thought, then are they worth belaboring?

All that being said, I can attest that the authors’ anatomical interpretations are well founded and accurate. Figures 1-6 are sufficiently detailed and accurate to say this. Figures 7 and beyond are not. As noted by the authors, the latter illustrations include a great deal of “artistic license”,  are quite inaccurate, ambiguous, and in my opinion not deserving of much speculation. This applies as much to the human as the ‘horse’. One thing is clear, however, the prominent third trochanter indicates that the horse strongly influenced the illustrations.

Below are some additional specific notes.

Line 45: incomplete sentence “…by the numerous [?] dedicated…”

Line 151-154 and figure 3: it might be helpful to point out whether the inset is a drawing of a dissection of the front or rear foot, as the terminology is ambiguous.

Line 189+ I seriously doubt that anyone could mistake RCIN 919097 (figure 6) “at a glance” as an illustration of a human considering the lobes of the liver as drawn.

The illustration of the branches of the aortic arch are ambiguous (figure 6) due to the quality of preservation. I could easily be convinced there is third vessel depicted that is unaccounted for. There are certainly at least two branches, definitely excluding the horse.

Lines 220-221: it is not clear to me whether the authors are implying that da Vinci modified his illustration of a dog to meet his expectations of a horse, thus misrepresenting a composite drawing. Indeed, if I understand correctly “This text refers to the drawing RCIN 919097-recto, entitled ‘The viscera of 188 a horse’ (1490-92)", then is was da Vinci who identified this as a horse. If it was not da Vinci, then the authors need to make this clearer.

Usage of right and left is inconsistent, e.g., lines 204 vs 225-226, as applied to subject or view of illustration.

Lines 336-337: it is somewhat absurd to suggest that the organ caudal to the heart could be anything other than the liver considering that the caudal vena cava terminates there. Having said that, the illustration of the liver is rather crude despite the fact that it is large and well defined. It is as consistent with dog as anything else, but it could conceivably be that of any number of different species.

Author Response

This manuscript is a detailed re-evaluation of the historical interpretations of a few of Leonardo da Vinci’s illustrations. I can understand the interest in a historical figure like da Vinci. However, it was not da Vinci who misidentified the bear’s foot as left rather than right, it was the interpretation of others centuries later. Since this is not a reflection on da Vinci but on others, then I wonder how significant this is even from a historical perspective. The instances of inaccuracies by prominent figures in age-old scientific literature are an innumerable. Unless they are misguiding current thought, then are they worth belaboring?

Response: We could be wrong, but we think that the only way to move forward is revisiting the known and unknown, questioning the previous statements, finally contributing to a trustful knowledge, not only for entertainment, but also for academic and research reasons. Hence, we believe it is worthy and a commitment with the demographic visiting the websites where these masterpieces are displayed.

We fully agree with the comments made by Reviewer-2 “These amendments will highlight the anatomical drawings by Leonardo on animal anatomy for wider audience, such as anatomists, veterinary academics, researchers, and those interested in the world of the ‘Renaissance man’ skilled and versed in arts and sciences.”

All that being said, I can attest that the authors’ anatomical interpretations are well founded and accurate. Figures 1-6 are sufficiently detailed and accurate to say this. Figures 7 and beyond are not. As noted by the authors, the latter illustrations include a great deal of “artistic license”, are quite inaccurate, ambiguous, and in my opinion not deserving of much speculation. This applies as much to the human as the ‘horse’. One thing is clear, however, the prominent third trochanter indicates that the horse strongly influenced the illustrations.

Response: We have chosen these drawings displaying the comparative anatomy between man and horse, because they are unique (to the best of our knowledge there are no more) and revisit them. These drawings show the huge interest of Leonardo in the comparative anatomy.

Reviewer-1 also says “The authors used comparative anatomy to stand out the knowledge on animal anatomy in the Renaissance, specifically five hundred years back. The manuscript amends some descriptions and reveals new undescribed features. Those findings honour the excellence of Leonardo da Vinci, not only as an artist, but also as an anatomist, even on the lest known field of animal science.”

Below are some additional specific notes.

Line 45: incomplete sentence “…by the numerous [?] dedicated…”

Response: The missing word is ‘sheets’: …’by the numerous sheets dedicated to his anatomical studies.’ (Line 99, tracked version; Line 48, clean version).

Line 151-154 and figure 3: it might be helpful to point out whether the inset is a drawing of a dissection of the front or rear foot, as the terminology is ambiguous.

Response: This fact was also commented by Reviewer-2, accordingly we have changed the terminology (from line 144 and onwards, tracked version), using the veterinary anatomical nomenclature, in the main body and in figures, but the title of some figures, to keep them as they were named in the inventory of the Royal Collection Trust.

Line 189+ I seriously doubt that anyone could mistake RCIN 919097 (figure 6) “at a glance” as an illustration of a human considering the lobes of the liver as drawn.

Response: The whole sentence was removed as it does not give extra information. Instead, we added a description of the drawing, as suggested by Reviewer-2: ‘The drawing represents the ventral aspect of the trunk of an animal (supposedly, a horse) with the lungs and canalis alimentarius (esophagus, stomach, and intestines) removed’ (Lines 293-294, tracked version; Lines 197-198, clean version). The description that appears at the Royal Collection Trust website [3] was also included (Lines 291-292, tracked version; Lines 195-196, clean version ),

The illustration of the branches of the aortic arch are ambiguous (figure 6) due to the quality of preservation. I could easily be convinced there is third vessel depicted that is unaccounted for. There are certainly at least two branches, definitely excluding the horse.

Response: At higher magnification in this screenshot, just to justify the vessels at the arcus aortae, we coloured the arteries in red. As it is shown, there are two main vessels (truncus brachiocephalicus and A. subclavia sinistra), and between the second one (A. subclavia sinistra) and the third one there is a striped area (blue arrow) that could represent the intercostal pleura and muscles. Then the third vessel could be the first intercostal branch (upper green arrow), and the other two small branches (green arrows) could be the second and third intercostal arteries.

Lines 220-221: it is not clear to me whether the authors are implying that da Vinci modified his illustration of a dog to meet his expectations of a horse, thus misrepresenting a composite drawing. Indeed, if I understand correctly “This text refers to the drawing RCIN 919097-recto, entitled ‘The viscera of 188 a horse’ (1490-92)", then it was da Vinci who identified this as a horse. If it was not da Vinci, then the authors need to make this clearer.

Response: The sentence (Line 221): ’fact that supports Leonardo knew the internal anatomy of the horse’ was removed in order to avoid any confusion.

In general, Leonardo da Vinci did not name most of his drawings (around 5000 sheets). These were itinerant (belonging to different owners) during more than two centuries. Later on, Charles II, King of Great Britain, got them by purchase or gift (around 1660-85), and since then, they are part of the Royal Collection, in charge of doing the inventory, assigning to each of them a name and an inventory/catalogue number. In addition, the Royal collection Trust website [3] includes this information about his provenance:Bequeathed to Francesco Melzi; from whose heirs purchased by Pompeo Leoni, c.1582-90; Thomas Howard, 14th Earl of Arundel, by 1630; probably acquired by Charles II; Royal Collection by 1690.”

Consequently, to avoid any confusion, this sentence was included in the manuscript (Lines 290-293, tracked version; Lines 195-197, clean version): …“and described at the Royal Collection Trust [3] as: ‘an anterior view of the arteries, veins and the genito-urinary system of an animal, probably a horse”, implying that Leonardo did not name this drawing.”

Usage of right and left is inconsistent, e.g., lines 204 vs 225-226, as applied to subject or view of illustration.

Response: This was solved at Line 312 (tracked version) or Line 202 (clean version), and onwards, using dexter/dextra and sinister/sinistra when referred to vessels or organs, and right and left when referred to the image.

Lines 336-337: it is somewhat absurd to suggest that the organ caudal to the heart could be anything other than the liver considering that the caudal vena cava terminates there.

Having said that, the illustration of the liver is rather crude despite the fact that it is large and well defined. It is as consistent with dog as anything else, but it could conceivably be that of any number of different species.

Response: Are you referring to lines 230-231?

Just looking at the liver, many species have similar lobulation. Hence, we have checked the major vessel represented comparing the domestic mammals. In consequence, the depicted details are more in accordance with the dog anatomy than with the horse anatomy.

Just in case, we have changed “Assuming this” for “Accordingly” (line 370, tracked version; line 243, clean version).

Thank you very much for your comments and suggestions. We hope you consider that our manuscript has improved.

Reviewer 4 Report

This paper is a brilliant review of the anatomical work based on Leonardo da Vinci's animal drawings. The authors have clearly defined the topic and carried out an extensive literature review and anatomical study. Leonardo's drawings were carefully analyzed, discussed and properly amended in some aspects regarding the bear and horse anatomy. Mention to dog and human anatomy is very well documented. Also, a continuous reference to previous works is used to support the discussion. All the criticism is very well supported by accurate knowledge of animal comparative anatomy. The conclusions are clear and robust.

The whole manuscript is very well organized, and remarkably easy to read. This achievement must be enhanced because writing an historical-review of this kind in a clear and comprehensive way is very challenging 

The paper is ready for publication. No minor changes are required.

Author Response

This paper is a brilliant review of the anatomical work based on Leonardo da Vinci's animal drawings. The authors have clearly defined the topic and carried out an extensive literature review and anatomical study. Leonardo's drawings were carefully analyzed, discussed and properly amended in some aspects regarding the bear and horse anatomy. Mention to dog and human anatomy is very well documented. Also, a continuous reference to previous works is used to support the discussion. All the criticism is very well supported by accurate knowledge of animal comparative anatomy. The conclusions are clear and robust.

The whole manuscript is very well organized, and remarkably easy to read. This achievement must be enhanced because writing an historical-review of this kind in a clear and comprehensive way is very challenging.

The paper is ready for publication. No minor changes are required.

ANSWER

We are very pleased you liked our manuscript and for the supportive and encouraging comments.

We do hope our revised manuscript also meets your approval.

Round 2

Reviewer 3 Report

Nothing in the authors' revisions or responses to my original review changes my conviction that the journal Animals simply is not an appropriate venue for this manuscript. This manuscript represents an anecdotal evaluation of others' interpretations of the work of a historical figure. It does not advance knowledge of any aspect of contemporary animal science. I see no reason why any animal science researcher, veterinarian, or animal husbandry practitioner would refer to this paper other than for casual amusement.